# Bone Loss in Distal Radial Fractures Treated with A Composite Xenohybrid Bone Substitute: A Two Years Follow-Up Retrospective Study

**DOI:** 10.3390/ma13184040

**Published:** 2020-09-11

**Authors:** Riccardo Ferracini, Alessandro Bistolfi, Claudio Guidotti, Stefano Artiaco, Agnese Battista, Bruno Battiston, Giuseppe Perale

**Affiliations:** 1Department of Surgical Sciences and Integrated Diagnostics, University of Genova, Viale Benedetto XV n6, 16132 Genova, Italy; 2Department of Traumatology and Rehabilitation, C.T.O. Hospital-A.O.U. Città della Salute e della Scienza, Via Zuretti 29, 10126 Turin, Italy; abistolfi@cittadellasalute.to.it (A.B.); sartiaco@cittadellasalute.to.it (S.A.); bbattiston@cittadellasalute.to.it (B.B.); 3Medical School, University of Turin, 10100 Turin, Italy; claudio.guidotti@edu.unito.it (C.G.); agnesebattista92@gmail.com (A.B.); 4Industrie Biomediche Insubri S.A., Via Cantonale 67, 6805 Mezzovico-Vira, Switzerland; giuseppe@ibi-sa.com

**Keywords:** xenograft, bone graft, radial fracture, xenohybrid biomaterial

## Abstract

(1) Background: Recently, surgical treatment of distal radius fractures has increased exponentially. Many locking plates’ fixation systems have been developed allowing a more stable reduction and early mobilization. Sometimes, open reduction and fixation of distal radius fractures may leave a residual bone loss requiring grafting. This retrospective study reports clinical and radiologic outcomes of distal radius fractures treated with xenohybrid bone grafting in order to assess (i) the safety of the investigated bone graft; (ii) its radiological integration and biomechanical performances, and (iii) clinical outcomes of the patients; (2) Methods: We performed a retrospective study on a cohort of 19 patients. Preoperative X-ray and CT scan were performed. The mean clinical and radiographical follow-up was two years. Safety of the xenohybrid bone graft was constantly evaluated. Clinical results were assessed through the DASH score and Mayo wrist score; (3) Results: No adverse reactions, infections, and local or general complication were related to the use of xenohybrid bone graft. The radiolucency of the xenografts suggested progressive osteointegration. No evidence of bone graft resorption was detected. All the patients reached consolidation with good to excellent clinical results; and (4) Conclusions: Clinical and radiological data demonstrated that xenohybrid bone grafting promotes new bone formation and healing in osteopenic areas caused by fracture reduction.

## 1. Introduction

Distal radial fractures account for 17% of all skeletal fractures [1,2,3], which makes them quite common. The vast majority of them occurs in male patients under 30 years old, mostly due to high-energy traumas, and, in over 60 year-old female subjects, mostly due to falls. A stable, congruent, well-aligned, and painless wrist joint along with a wide range of motion are of paramount importance during the healing process, in order to avoid potentially severe complications like e.g., post-traumatic osteoarthritis and stiffness. The volar approach and fixation with locked plate and screws is now of widespread use because open reduction internal fixation (ORIF) guarantees both an anatomical reduction of the fracture and best results from a clinical and functional point of view [4].

The use of bone graft in surgical treatment of distal radius fractures is still discussed and controversial. Nonetheless, it is gaining popularity, especially in patients with comminute fractures or excessive bone loss caused by traumatic trabecular collapse, bone graft can be useful to support bone fixation. This may occur in osteoporotic bone or impacted osteoarticular fragments that have lost metaphyseal support, and in dorsally plated comminute distal radius fractures [5,6,7,8,9].

Bone grafting scaffolds were thoroughly studied in recent years, demonstrating advantages and limitations [10,11]. Ideally, a bone substitute should demonstrate the following properties: osteoconductivity and osteoinductivity, along with vascular ingrowth and bone remodeling, and the ability to stimulate mesenchymal stem cells (MSCs) to differentiate into osteoblast progenitors [12]. The current standard of choice for bone grafting is represented by autologous iliac bone graft (AIBG); however, there are some restrictions to this technique, such as donor site morbidity and the needs for an additional surgical approach, both of which are relevant [13,14,15]. Bone allografts, from live donor or cadaver, were also considered, though they carry a potential risk of disease transmission and biocompatibility which can impact clinical outcomes [16]. Hence, synthetic bone grafts are gaining popularity among alternative strategies, since they are easily customizable, readily available off the shelf and with most of the abovementioned properties present [17]. One of the major issues is represented by biocompatibility and biomechanical stability, which are addressed by the use of calcium phosphates (CP) materials. Nonetheless, recent animal studies showed that CP-containing grafts are reabsorbed and remodeled too fast, in only 12 to 26 weeks [18,19,20].

Another valuable form of bone graft scaffold is represented by bovine xenografts, which very accurately mimic the structure of human cancellous bone [21]. Still, they are not free from major issues such as sterilization and the need to improve the biomechanical properties [22,23]. SmartBone^®^ (SB) (Industrie Biomediche Insubri S.A., City, Switzerland) is a composite xenohybrid scaffold (CXS), obtained from a bovine bone-derived matrix, reinforced with poly(L-lactide-co-ε-caprolactone) (PLCL) and RGD-containing collagen fragments (obtained from animal-derived gelatin), which improve elasticity, blood affinity, and cell attachment, respectively, and proved to be a robustly reliable bone graft in serval regeneration indications overall [24,25,26,27,28].

Regarding the possible interaction of biomaterials with local tissue, various studies emphasized how the behavior of stem cells can be modulated (i.e., inflammation, angiogenesis, and bone regeneration) [29,30].

In this work, part of a wider clinical study, we present clinical and radiological findings of a non-consecutive series of patients with radial wrist fractures treated with open reduction and internal fixation (ORIF) augmented with the CXS described above. The aims were to: (1) assess the mid-term safety of the investigated bone graft; (2) its radiological integration and biomechanical performances, and (3) clinical outcomes of the treated patients over a two-year timeframe.

## 2. Materials and Methods

### 2.1. Study Design, Patients’ Selection, and Endpoints

Within the framework of a wider clinical study, we performed a retrospective analysis of collected database of patients treated for distal radius fracture. The study was based on clinical data and radiological images prospectively recorded before, during, and after the surgical procedure. Regular clinical and radiological checks were performed, and, finally, a patient self-reported disability score (DASH) and a functional score (Mayo wrist score) were used to assess at two years of follow up.

We considered a cohort of patients treated for distal radius fracture from June 2016 to June 2018. Inclusion criteria to participate in this study were: age ≥ 18; patients who expressed their informed consent to participate the study; diagnosis of radial wrist fracture; bone augmentation with SB; and availability of complete clinical and radiological data from follow-up. Exclusion criteria were age < 18 and the presence of comorbidities, such as e.g., metabolic bone diseases, diabetes, or malignancies. Gender or tobacco use were not considered as exclusion criteria.

Nineteen patients met the inclusion criteria and were included in the study, 9 males (mean age 47 years, std. 13) and 10 females (mean age 62 years, std. 13). The mean age of this group of patients was 55 (std. 15) years, ranging from 30 to 80 years. The right wrist was involved in 12 cases and the left wrist in the remaining 7.

The main endpoints of the study were: (a) to detect the possible adverse effects or complications of the performed procedures; (b) to evaluate the xenograft integration with the surrounding bone and the ability to promote bone matrix production, through the mid-term radiographic evaluation; (c) to evaluate the clinical outcomes.

All patients had surgery under general anaesthesia; a tourniquet was routinely positioned before starting the surgical procedure; all patients were operated in a supine position, with the injured arm on a surgical table. Fractures were reduced through a volar approach, and fixation was performed with volar plates and screws. SB blocks were customized to the defect during surgery, starting from commercially available blocks. The shaped blocks were positioned considering the volume of the bone defect for each patient (Figure 1). After surgery a splint was applied for four weeks, the splint was further removed, and the rehabilitation exercises started. The decision to use a bone substitute was taken during the surgical procedure when a relevant bone gap was seen in fractured bone and/or when impacted osteoarticular fragments lost their metaphyseal bone support.

### 2.2. Follow-up: Radiographic Examination

All the patients performed radiographic examinations during regular follow-up. Radiographs were examined for callus formation, bone resorption, implant stability, and SB integration [28] (Figure 1, Figure 2 and Figure 3).

### 2.3. Follow-up: Clinical Examination

Full wrist examination was performed during clinical checks recording range of motion (ROM) of the wrist in flexion–extension and in pronation–supination. Clinical signs of infection and neurologic and/or vascular impairment were considered. During the final follow-up, we investigated the results according to the Mayo wrist score and Disabilities of the Arm, Shoulder, and Hand (DASH) score [30,31,32].

The Mayo wrist score considered residual pain, functional status, range of motion, and grip strength compared to the contralateral side. Finally, the disability of the upper limb was evaluated with the DASH score. The outcome measure is a 30-item, self-report questionnaire designed to assess the patient’s health status during the previous week previous to evaluation. The items inquire about the degree of difficulty in performing different physical activities because of arm, shoulder, and hand problems (21 items), the severity of each of the symptoms of pain, activity-related pain, tingling, weakness, and stiffness (five items), and the impact of the problem on social functioning, work, sleep, and self-image (four items). Each item has five response options. The scores are then used to calculate a scale score ranging from 0 (no disability) to 100 (most severe disability).

### 2.4. Ethical Compliance

This study is part of an observational study protocol that is sponsored by I.B.I. S.A. and was approved by the United Ethical Committee of the “Città della Salute e della Scienza”, Turin, Italy (approval No. 0004336), which includes the hospital where all patients were treated. All patients signed an informed consent form to document that they understood the aims of the study and authorized the use of their data for research purposes. This study was performed according to the International Ethical Principles following the recommendations of the Declaration of Helsinki as revised in Fortaleza (2013) for investigations with human subjects and followed good clinical practice.

## 3. Results

### 3.1. Radiological Assessment

The fractures were classified according to the AO Classification: most cases were 23 C1 or 23 C3 fractures. Among them, we treated one case of 23 A2 and one case of 23 B3 fractures.

X-ray images of 7 out of 19 patients showed that fractures were consolidated at two months: the radiographic controls, indeed, showed abundant callus formation. An exemplificative case is presented in Figure 2. The remaining 12 patients showed reduced callus formation after two months and performed an additional X-ray imaging check at five months after surgery (exemplificative case shown in Figure 3). In all cases, radiolucency of the xenografts was assessed, which became progressively more similar to that of the surrounding healthy bone tissue.

### 3.2. Clinical Assessment

All the patients completed the follow up study. No infection, neurologic, and vascular complications, or xenograft implant-related adverse effects, were reported during postoperative controls. Three patients underwent hardware removal for residual local pain and to tendon inflammation. The patients had plate removal due to hardware intolerance, but they showed a good intraoperatively evaluated stability of the SB implant and therefore completed either the Mayo or the DASH questionnaires.

All the patients performed physiotherapy for at least one month. The recovery of the wrist ROM was progressive, with a mean flexion of 30° (±10°) and extension of 30°(±10°) from the starting neutral 0° (±10°) and a pronation of 50°(±20°) and a supination of 50° (±20°) from the startin neutral 0°at three months. Most of the patients had a complete recovery of the ROM at nine months. Prolonged physiotherapy was continued in the patients with a slow progressive ROM recovery. The DASH score was used for the subjective evaluation of patients’ disability. The average score obtained was 9. The highest score was 36.6 and the lowest score was 1.5. Table 1 reports the results of the DASH score while Table 2 reports the functional results at two-year follow-up according to the Mayo wrist score (avg. score 80 std. 15; minimum score 45; max score 100).

## 4. Discussion

Several strategies for the treatment of distal radius fractures have been described in the literature including closed reduction and splinting or casting, external fixation, and open reduction and internal fixation. Although both conservative and surgical management have been reported to be successful, current evidence and the most recent American Academy of Orthopaedic Surgeons (AAOS) clinical guidelines comparing conservative and surgical treatment of distal radius fragility fractures are inconclusive [33]. Although cast immobilization alone avoids the potential complications of surgery, it may fail in maintaining the correct reduction, and it is also associated with a certain risk of late fracture collapse and malunion; therefore, most physicians advise surgical fixation because it improves stability and enhances earlier recovery of ROM [34].

In some cases, the bone loss occurring in fragmented displaced fractures of some patients, and/or the severe collapse of bone fragments of the articular surface, can be reasonably treated with bone grafts or bone substitutes.

The most popular kind of bone graft is cortico-cancellous autograft obtained from the iliac crest, distal femur, proximal tibia, fibula, distal radius, and olecranon [35].

Recently, Suda et al. observed that that donor site morbidity after harvesting a small quantity of iliac crest bone graft for distal radius fractures is lower than the morbidity published for harvesting larger grafts for other indications. Nonetheless, they reported some cases of hematomas and one re-operation for bleeding from iliac crest [36].

Harvesting bone graft from the iliac crest is a reliable procedure, but it may cause complications like pain, bleeding, infection, and nerve injuries. Furthermore, prolonged operative time, limitations to the quantity of bone that can be safely harvested, and the risks of donor site morbidity are still concerning factors which led to the use of bony allograft and to the development of bone substitutes [13].

Compared to bone graft, bone substitutes are easily storable in the operating theater and readily available in different shapes and sizes for immediate use during surgery.

Among the bone substitutes, the use of SB was supported by previous studies in several anatomical sites and with different indications [26]. On this basis, we used SB in selected cases of distal radius fracture when we considered it advisable to fill bone defects in osteopenic patients or to support articular fragment in comminuted displaced fractures. In this study, radio-transparency reduction of SB graft over time was used as an indirect measure of bone graft mineralization by host, evidencing the occurrence of the same anatomically selective remodeling mechanism that has been already seen and described in other anatomical districts such as e.g., tibial plateau, oral and maxillo-facial spaces [23,25,27,28].

Clinical outcome scores were classified as good to excellent, given that only two patients reported average (45 and 55) scores, mainly because of the severity of the injury that caused the displaced fracture. Similar results were achieved by Cano-Luis et al. regarding post-traumatic bone defects [37].

As highlighted by Marrelli et al. [38], pre-clinical investigations are still of paramount importance. Extensive research was done about graft choosing strategy and about the appropriateness of biomechanics achieved by the final graft [39,40]. Similarly, Bracey et al. discussed the in vitro safety of bone xenografts, demonstrating human biocompatibility and biologic safety [41]. Wide in vitro studies were performed on SB too, confirming not only its safety, but also its mechanism of action also from a cell biology perspective [12], perfectly matching clinical and histological evidence [23,24,25,26,27,28].

This study showed some limitations, as the small number of non-consecutive patients, the retrospective nature of the analysis, which includes different subtypes of fractures, and the absence of a control group using other bone graft approaches, which was referred to literature given experiences. Despite this, the endpoints of the study have been completely documented. At final follow-up, the xenohybrid bone substitute demonstrated being safe and biocompatible and showed adequate mechanical properties to sustain healing and consolidation of distal radius fractures. Radiological follow-up did not show diastasis or depressions of articular surface, meaning that mechanical support of the plate and screw fixation with SB adjunct were adequate for high and complex forces, like those which commonly stress the wrist. X-ray controls showed that the radiolucency of implants was progressively more similar to radiolucency of the surrounding bone. As in previous clinical investigations, we interpreted this finding as SB integration with autologous bone tissue and deposition of new bone matrix within an ongoing remodeling process leading to the formation of new living bone. Clinical results were largely positive showing fracture healing and good or excellent functional recovery in most of the patients according to a Mayo wrist score. No infection, neurovascular complications, or implant-related adverse effects were reported during follow-up.

## 5. Conclusions

Innovative biomaterials for bone substitution have demonstrated over time to be safe, reliable, and biocompatible tools for orthopedic surgery [21,26]. Their use may reduce complications, operative times, operative costs, and hospital length of stay [13]. In our clinical series, out of 19 patients, radiological and functional outcomes were overall positive, ranging from good to excellent scores. Moreover, patients were satisfied with the ultimate result. No complications or reactions linked to the material have been detected. At the final follow-up, the xenohybrid material investigated in our study has shown to be a safe and useful resource for orthopedic surgeons in the treatment of distal radius fractures, thanks to its biocompatibility and capacity of osteointegration. This evidence suggests that SmartBone^®^ can be considered for orthopedic surgery when bone graft is required and also in the forearm segment.

## Figures and Tables

**Figure 1 materials-13-04040-f001:**
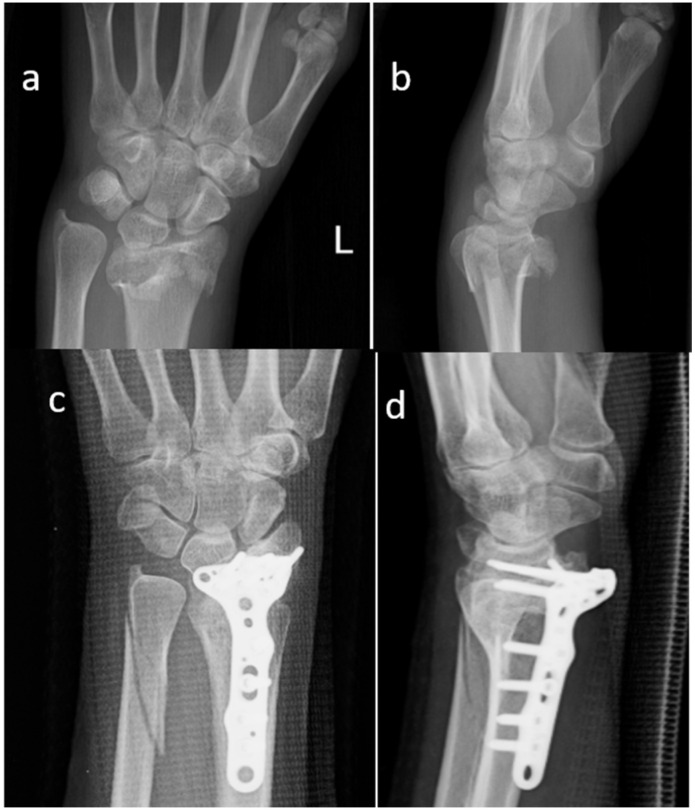
Standard X-ray of a displaced articular fracture of the distal radius associated with multifragmented distal ulnar fracture. Preoperative images in AP (**a**) and Lateral (**b**) view are compared to post-operative X Ray at 2 months (**c**,**d**). The osteosynthesis appears stable and the xenograft block appears stable. (Female, 34 yy).

**Figure 2 materials-13-04040-f002:**
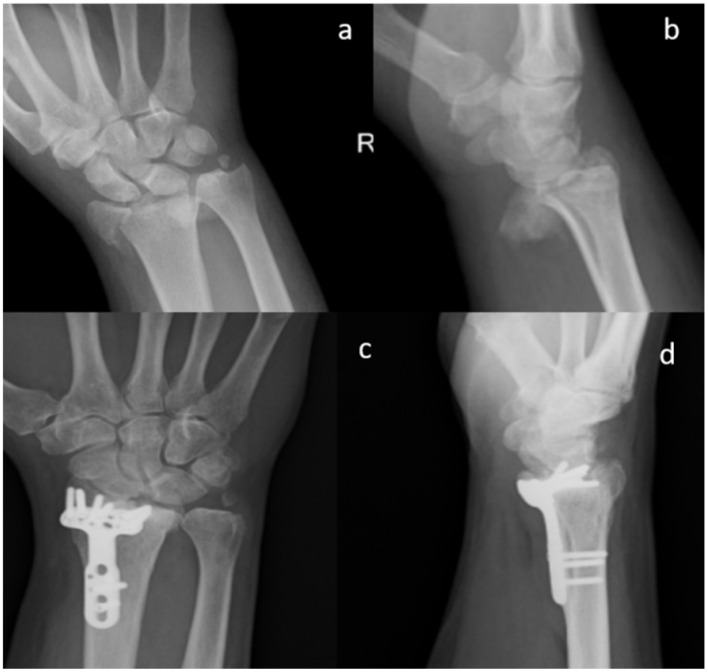
Standard X-ray of a displaced articular fracture of the distal radius associated with styloid ulna fracture. Preoperative images in AP (**a**) and Lateral (**b**) view are compared to post-operative X Ray at 2 months (**c**,**d**). The exuberant callus formation can be visualized (Male, 55 yy).

**Figure 3 materials-13-04040-f003:**
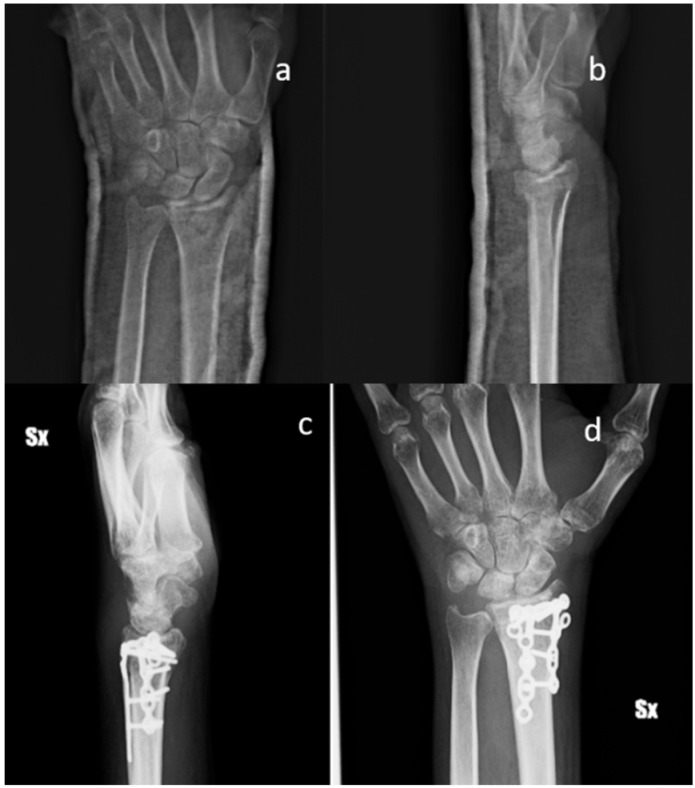
Standard X-ray of a displaced articular fracture of the distal radius associated with styloid ulna fracture. Preoperative images in AP (**a**) and Lateral (**b**) view are compared to post-operative X Ray at five months (**c**,**d**). The images show callus formation and remodeling of the smart bone. (Male, 50 yy).

**Table 1 materials-13-04040-t001:** Reports the AO Classification of the fractures, the clinical results at the final follow-up and the Disabilities of the Arm, Shoulder and Hand (DASH) at two years follow-up. AO = AO classification of the fracture. ROM = range of movement; when similar to the contralateral uninjured wrist the ROM is reported as “complete”; limitations are described: slightly limited = reduction less than 10 degrees, limited reduction from 10 to 20 degrees. Pain is reported according to the NRS (Noise reporting Scale) from 0 to 10. Deficit column reports the neurological, sensorial, or motor limitations and are described each case; NO = no deficit reported.

AO	ROM	Pain	Deficit	DASH
2R1C1	Complete	0	paraesthesia thumb and index	6.7
2R1C1	Complete	0	NO	3.3
2R1C3	slightly limited (only supination)	0	NO	5.8
2R1C1	Complete	0	NO	1.6
2R1C1	Limited	0	NO	10.8
2R1C1	slightly limited	0	NO	5
2R1C3	Limited (difficult fingers flexion)	0	difficult fingers flexion for scar adherences	36.6
2R1C3	Complete	0	Arm paraesthesia	4.1
2R1C3	Limited (only extension)	3	NO	7.5
2R1C1	Complete	0	NO	2.4
2R1C1	Complete	0	NO	1,7
2R1B3	Limited (flexion-extension)	4	NO	6.6
2R1C1	Complete	0	NO	20.8
2R1C1	Limited	5	NO	17.5
2R1A2	Complete	0	Index paraesthesia	9.1
2R1C3	slightly limited	4	NO	20.6
2R1C3	slightly limited (only supination)	0	NO	7.8
2R1C1	Complete	0	NO	1.5
2R1C3	Complete	0	NO	2.6

**Table 2 materials-13-04040-t002:** Reports the Mayo WRIST Score at two years follow-up. Pain: Reported as a subjective patient description; Functional Status: Set accordingly to the patient’s ability to work, **RTRE =** Returned to Regular Employment; ROM 3a: Range of Motion compared to contralateral side; ROM 3b: Range of motion of the injured hand alone; Grip Strength: Evaluated clinically. **Pt**: patient. 1 Deficit due to displaced fracture not reducible. 2 patient who had second surgery and hardware removal.

Patients	Sex	Age	Total Score	(1) Pain	(2) Functional Status	(3a) ROM	(3b) ROM	(4) Grip Strength
Pt 1	F	80	**100**	No pain	RTRE	100%	>120°	100%
Pt 2	M	39	**85**	Mild, Occasional	RTRE	75–99%	>120°	75–99%
Pt 3	M	31	**90**	No pain	RTRE	75–99%	90–120°	100%
Pt 4	F	64	**85**	Mild, Occasional	RTRE	75–99%	>120°	75–99%
Pt 5 ^2^	F	62	**85**	Mild, Occasional	RTRE	100%	>120°	75–99%
Pt 6	M	30	**100**	No pain	RTRE	100%	>120°	100%
Pt 7	F	73	**60**	Moderate, Tolerable	Able to work, Unemployed	100%	>120°	25–50%
Pt 8	F	54	**70**	No pain	Able to work, Unemployed	75–99%	90–120°	75–99%
Pt 9 ^1^	M	55	**45**	Mild, Occasional	**Restricted ^1^**	0–24%	<30°	25–50%
Pt 10	F	56	**80**	No pain	RTRE	75–99%	90–120°	75–99%
Pt 11	M	49	**85**	Mild, Occasional	RTRE	75–99%	>120°	75–99%
Pt 12	F	74	**65**	Mild, Occasional	Able to work, Unemployed	75–99%	90–120°	75–99%
Pt 13 ^2^	M	49	**90**	No pain	RTRE	75–99%	90–120°	100%
Pt 14	M	59	**100**	No pain	RTRE	100%	>120°	100%
Pt 15	F	75	**55**	Mild, Occasional	Able to work, Unemployed	50–74%	60–90°	50–74%
Pt 16	M	73	**65**	Mild, Occasional	Able to work, Unemployed	75–99%	90–120°	75–99%
Pt 17 ^2^	F	50	**85**	Mild, Occasional	RTRE	75–99%	>120°	75–99%
Pt 18	M	37	**90**	No pain	RTRE	75–99%	90–120°	100%
Pt 19	F	34	**90**	No pain	RTRE	100%	90–120°	75–99%

^1^ Deficit due to displaced fracture not reducible. ^2^ Patient who had second surgery and hardware removal.

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
