# Peer review of "Bone Loss in Distal Radial Fractures Treated with A Composite Xenohybrid Bone Substitute: A Two Years Follow-Up Retrospective Study"

_materials, 2020, doi:10.3390/ma13184040_

Round 1

Reviewer 1 Report

The topic of this article entitled “Bone loss in distal radial fractures treated with a composite xenohybrid bone substitute: a two years follow-up retrospective study.” is aimed to assess the relationship between bone loss and novel bone substitutes. It is an interesting topic and within the journal's scope. Nevertheless, this reviewer would suggest some improvements, before further considerations.  The study has certainly new information related to the role of bone repairing in presence of different conditions and biomaterials. This article also aims to translate the main key-concepts to tissue engineering and translational medicine in the very next future.

The main strength of this paper is related to the appropriateness of approach, and to the study design, which were both really relevant. Moreover, the soundness of the whole article and the relevancy of discussion have merits, as well. The clarity of writing is also fine; however, some key-concepts should be slightly improved for a major clarity to the readers, involving other critical discussion and better comparing some evidence reported in previously published papers.

Introduction: Authors have reported several important topics related to tissue healing and repairing; however, poor has been reported on the other resident stem cells, which can act as the immunomodulatory and pro-osteogenic activities in the local environment.

Authors have consistently discussed on the role of tissue engineering. However, something more should be discussed about the role of specific “biomaterials” or “scaffolds” as study model. In this light it’s important to briefly describe something about the “safe” in-vitro reparative models, working without any additive (e.g. BSA) to apply safely such protocols in human models. Finally, the strategies to chose materials and manage loads should be briefly described in the discussion section, highlighting the role of pre-clinical investigations on this matter (Please, see and discuss “Marrelli M, Maletta C, Inchingolo F, Alfano M, Tatullo M. Three-point bending tests of zirconia core/veneer ceramics for dental restorations. Int J Dent 2013; 2013, 831976.”).

- Conclusions should be improved with clear take-home messages.

Minor suggestion:

- In the whole text there are some typos here and there: authors should carefully revise the text before resubmission.

Author Response

- In the discussion section, we cited specific articles that mentioned in-vitro studies on SmartBone, not only regarding safety of the biomaterial but also its role on biomechanical properties and on its mechanism of action from a cell biology perspective.  The same was done about the choosing strategy of the material and the importance of the pre-clinical investigation. Please see the full text for highlighted and bold text regarding revisions. You can also find the updated bibliography about SmartBone investigations (Cingolani et al 2018).

- We accept the comment on the conclusion section which was slightly redacted as requested. 

- Unfortunately, regarding the role of local stem cells, we believe that the current study does not aim to discuss the immunomodulatory activities of the local environment, hence we will reserve that topic for another study, however we have added a comment and a reference on SmartBone in vitro investigations involving stem cells to describe its mechanism of action (see also ref 40).

- English check was performed and typos were corrected  (but not highlighted for sake of easiness in reading)

Author Response

- Minor mistakes were addressed as requested (but not highlighted for sake of easiness in reading)

- Regarding the osteoporotic bone, we are sorry for the oversight. With "osteoporotic" we intended the poor quality of the residual fractured bone. It was appropriately corrected. 

Reviewer 3 Report

The current manuscript "Bone loss in distal radial fractures treated with a composite xenohybrid bone substitute: a two years follow-up retrospective study" is a restrospective analysis of bone graft material functional performance using two different scoring systems. The manuscript needs major revisions regarding data presentation and discussion. The specific comments are listed below-

  1. Abstract- Xenohybrid name and composition should be mentioned in the abstract. The authors can't conclude- "Clinical and radiological data demonstrated that xenohybrid bone grafting promotes new bone formation and healing in osteopenic areas caused by fracture reduction" as there is no postoperative radiological data showed in the current manuscript.
  2. No description of how the bone graft was used. Representative images should be included with description of the graft size/ volume/ defect size for patients in a tabular form.
  3. In the Results section, Representative radiological images of post-operative bone healing must be included in the results. The readers can't comprehend the conclusions without these.
  4. Section numbering is wrong in the results section.
  5. Demographics is a part of methods not results, should be moved.
  6. Results section needs rearrangement, figures and tables should be embedded in the corresponding results.
  7. No discussion of the score results from other studies. This study is only based on scoring which has not been discussed at all.
  8. Conclusion- "Xenohybrid material investigated in our study has been a safe and useful resource for orthopaedic surgeons in the treatment of distal radius fractures."- This is a vague conclusion. What do authors mean by "safe and useful". There is no biocompatibility or other such studies performed in the current manuscript.

Author Response

- Xenohybrid name and composition were mentioned in the introduction section (citations 22-23), but due to the long names of the molecules and additives, we couldn't manage to insert it into the abstract without greatly impacting the 200 words limit.  
- We briefly described the insertion of the graft and we added the initial dimensions of the SmartBone blocks. Unfortunately, since the xenograft is modeled during surgery specifically for each patient defect, it was impossible to precisely report even average implanted dimensions. 

- Post-operative radiological images are present in the paper (Figures 1, 2 and 3). Please see the text for highlighted parts in Figures captions.
- We corrected section numbering and the demographics paragraph was moved to Section 2 as requested.

- We briefly discussed results and we cited similar results extracted from the recent literature

- As mentioned above, we rearranged the conclusion section

Reviewer 4 Report

The analysis of bone loss in distal radial fractures treated with composite xenohybrid bone substitute showed significant improvement in new bone formation and healing in the osteopenic area caused by fracture reduction. Though, it's very relevant clinical study but provides good information. I will recommend it for publication with minor revision. 

Here authors have collected most of the data using X-ray or CT scan, however they suggest that there is reduction in inflammation. What is the way they come to make this statement?

Can you please emphasize other relevant materials in introduction part, such as (Acta Biomaterialia 108 (2020) 97–110 and ACS Biomater. Sci. Eng. 2016, 2, 4, 454–472, etc)?

Most of the data are analytical like pain, inflammation, ROM, etc, is it possible to include experimental data related to inflammation?

Author Response

In this paper, our purpose was not to directly discuss inflammation and immunomodulatory activities of the local tissue surrounding SmartBone, instead we wanted to highlight the formation of callus on radiological images as a relevant clinical finding.

As mentioned by the reviewers, the presence of a control group would have been surely beneficial, but as of this moment we cannot comply with this request 

 as this paper is part of a retrospective clinical study where control data are taken from literature reported experiences.

Round 2

Reviewer 1 Report

authors have enough improved their paper. 

Reviewer 3 Report

The concerns have been addressed by the authors. The manuscript can be accepted in the current form.